# Gut Microbiota: An Immersion in Dysbiosis, Associated Pathologies, and Probiotics

**DOI:** 10.3390/microorganisms13051084

**Published:** 2025-05-07

**Authors:** Valentina Origüela, Alvaro Lopez-Zaplana

**Affiliations:** 1Department of Physiology, Faculty of Biology, University of Murcia, 30100 Murcia, Spain; valentina.origuelac@um.es; 2R&D Department, 3A Biotech, 30565 Murcia, Spain

**Keywords:** gut microbiota, probiotics, dysbiosis, intestinal diseases, gut–brain axis, metabolic disorders

## Abstract

The importance of the microbiome, particularly the gut microbiota and its implications for health, is well established. However, an increasing number of studies further strengthen the link between an imbalanced gut microbiota and a greater predisposition to different diseases. The gut microbiota constitutes a fundamental ecosystem for maintaining human health. Its alteration, known as dysbiosis, is associated with a wide range of conditions, including intestinal, metabolic, immunological, or neurological pathologies, among others. In recent years, there has been a substantial increase in knowledge about probiotics—bacterial species that enhance health or address various diseases—with numerous studies reporting their benefits in preventing or improving these conditions. This review aims to analyze the most common pathologies resulting from an imbalance in the gut microbiota, as well as detail the most important and known gut probiotics, their functions, and mechanisms of action in relation to these conditions.

## 1. Introduction

In a 70 kg human, the number of bacteria is approximately 3.8 × 10^13^, while the number of human cells is estimated at 3 × 10^13^ [1]. This makes the human body a true ecosystem where approximately 500–1000 bacterial species, with an estimated 2000 genes per species, coexist in symbiosis with our cells as a result of more than 500 million years of co-evolution [2,3]. The set of microorganisms that coexist in balance with our body is known as the microbiome. The microbiome plays a crucial role in host health by protecting against pathogenic microorganisms, modulating the immune response, contributing to neurotransmitter production, and participating in digestive processes such as fiber degradation [4].

The microbiome of a specific body part is referred to as the microbiota, and depending on its location, certain types of microorganisms will predominantly thrive. Thus, in the same person, the bacteria present on the skin, in the mouth, or in the intestine will not be the same, though they can be similar between different persons [5]. Additionally, the microbiome is unique to each individual and depends on factors such as genetics, age, gender, hygiene habits, stress, lifestyle, contact with nature, antibiotic use, or diet, among others [6]. Specifically, gut microbiota has a significant impact in human health, affecting nutrient absorption and influencing immune system or oxidative stress; in fact, it has been associated with metabolic syndrome and obesity [7,8,9].

A prolonged disturbance or imbalance in the microbiota can lead to dysbiosis, which is associated with various diseases [10]. In the mouth, dysbiosis can cause dental problems such as periodontal disease or the emergence of cariogenic bacteria like *Streptococcus mutans* [11,12]. In other organs, such as the intestine, dysbiosis can entail significantly more negative aspects, including digestive conditions such as colitis [13], or an increased risk of cardiovascular diseases [14] and neurological disorders [15]. Thus, these links between dysbiosis and disease are particularly evident and severe in the gut, where dysbiosis is associated with conditions such as inflammatory bowel disease (IBD) [16,17,18], metabolic disorders like obesity or diabetes [19], and various immune [20], neurological [21], and cardiovascular disorders [14]. In fact, the relevance of gut microbiota imbalances extends beyond chronic and metabolic conditions. Emerging evidence also highlights its role in acute infectious diseases, such as COVID-19. SARS-CoV-2 infection has been associated with significant alterations in gut microbial composition, reduced diversity, and increased intestinal inflammation, suggesting a broader systemic impact of dysbiosis in disease severity and immune response modulation [22].

To prevent such imbalances, we can consume prebiotics, substances that serve as nourishment for our beneficial microorganisms, or probiotics, non-pathogenic live microorganisms that offer certain health benefits [9,23]. This term was introduced by Élie Metchnikoff over 100 years ago, where he proposed the theory that manipulating the composition of the intestinal microbiota could benefit health [24]. The mechanisms by which probiotics inhibit the growth of other pathogenic bacteria and benefit us are diverse and depend on the specific probiotic strain. However, in general, they act by modifying the pH of the environment, producing antibacterial compounds and bacteriocins, competing for available nutrients and growth factors, or stimulating the host’s immune system [25,26]. Furthermore, to prevent the proliferation of other bacteria and generating dysbiosis, probiotics offer other health benefits, such as metabolizing indigestible fibers, producing vitamins and cofactors, promoting the production of anti-inflammatory cytokines and T-cell activity, and supporting intestinal barrier integrity, among others [26].

Due to the importance of the gut microbiota and its strong association with various pathologies, this review will examine the key relationships between gut microbiota and diseases linked to its dysbiosis, with an emphasis on how probiotics could improve or prevent the onset of these conditions.

## 2. Gut Microbiota and Its Modulation by Probiotics

In humans, gut microbiota weigh approximately 2 kg, with most of them being symbionts [27]. The gut microbiome maintains a state of homeostasis in our body that prevents pathogen colonization, influences intestinal permeability, facilitates the metabolism of certain foods such as fibers and specific types of sugars, synthesizes vitamins like K, B-complex, and folate, and modulates the local immune response while also influencing systemic immunity [28].

The human gut microbiota is predominantly composed of four major phylum: Bacillota (old Firmicutes) (including genera such as *Clostridium*, *Lactobacillus*, *Streptococcus*, *Enterococcus*, and *Eubacterium*), Bacteroidota (old Bacteroidetes) (*Bacteroides*, *Parabacteroides*, and *Provotella*), Actinomycetota (old Actinobacteria) (*Bifidobacterium* and *Collinsella*), and Pseudomonadota (old Proteobacteria) (*Helicobacter* and *Escherichia*). Additionally, two less-abundant phylum, Verrucomicrobiota (old Verrucomicrobia) (*Akkermansia*) and Fusobacteriota (old Fusobacteria) (*Fusobacterium*), are also present. Collectively, Bacillota and Bacteroidota account for approximately 70 to 90% of the gut microbial population [29,30,31]. There are some differences between the small intestine (duodenum, jejunum, and ileum) and large intestine (colon). The first group includes bacteria such as *Enterococcus*, *Lactobacillus*, *Bacteroides*, *Bifidobacterium*, *Clostridium*, and *Enterobacteriaceae*, while the second group comprises the same bacteria as the first, in addition to others such as *Escherichia*, *Klebsiella*, *Peptococcus*, *Peptostreptococcus*, *Proteus*, *Staphylococcus*, and *Ruminococcus* [32].

This standard microbiota can be modulated by prebiotics and/or probiotics. When probiotics are consumed in adequate amounts, they can colonize different parts of the digestive tract, protect the host from pathogenic microorganisms, and provide direct health benefits [10,26]. The world of probiotics is vast, and it is in continuous growth. Evidence of this is seen in an increasing volume of research and connections providing study of the gut microbiota and probiotics, which are closely linked to gastrointestinal disorders, immune diseases, and metabolic alterations, among others (Figure 1).

All microorganisms considered to be probiotics must possess a series of characteristics, as follows: they are live microorganisms which, when administered in adequate amounts, confer a health benefit on the host; they must be able to withstand gastric and bile acids, colonize a specific location within the digestive tract, mainly the large intestine, produce compounds that are beneficial to our health or directly provide some health benefit, and inhibit the growth of pathogenic microorganisms [33,34,35,36]. Among the main health benefits of probiotics are the enhancement of the immune system and reductions in the risk of cardiovascular, metabolic, or neurological disorders, which will be further developed in the following sections.

The two main genera of probiotic microorganisms are *Lactobacillus* and *Bifidobacterium* [37]. Other microorganisms less commonly considered are *Enterococcus*, *Streptococcus*, *Lactococcus*, *Pediococcus*, *Propionibacterium*, and yeast like Saccharomyces (*S. boulardii*, and *S. thermophilus*) [25,38]. For more detailed species and benefits of bacterial probiotics, see Table 1. In general, *Lactobacillus* and *Bifidobacterium* strains may improve metabolic outcomes by controlling glycemic levels, modulating the immune system, or inhibiting the destruction of pancreatic β-cells, thereby helping to prevent diabetes [39,40,41].

The mechanisms of action of probiotic bacteria are diverse. Many of these microorganisms produce metabolic byproducts such as acetic, propionic, and lactic acids, which contribute to a reduction in intestinal pH, thereby inhibiting the growth of pathogenic bacteria that usually prefer a neutral pH [120,121,122]. Probiotics can also physically limit adhesion sites, thereby preventing the attachment of pathogenic microorganisms, or directly compete with them for available nutrients in the intestine [123]. In addition to this physical exclusion, probiotics may exert biochemical inhibition through the synthesis of antimicrobial compounds or bacteriocins, which can hinder the proliferation of certain types of bacteria, including potentially pathogenic species [124]. Moreover, probiotics confer health benefits beyond the inhibition of pathogenic microorganisms. These include the production of certain bioactive compounds like vitamins (B6, B9, B12, or K), enzymes that enhance metabolic activity in the gut (proteases, lipases, amylases, or lactases), and amino acids and peptides that offer antioxidant and anti-inflammatory effects, as well as short-chain fatty acids (SCFAs) [26]. Notably, probiotics exhibit strong immunomodulatory activity by suppressing pro-inflammatory cytokines [125]. Some of these probiotic microorganisms can even inhibit the production of pathogenic enzymes that catalyze the conversion of carcinogenic precursors into carcinogens [126].

Currently, there is ongoing research into the use of engineered probiotics that could act as ‘sense and response’ systems (biosensor and biotherapeutic) [127]. These engineered probiotics would be bacteria that respond to specific biomarkers of inflammation by producing a therapeutic molecule, typically through transfected plasmids encoding for immunoregulatory cytokines or anti-inflammatory mediators, which are activated only upon induction of specific promoters [127,128]. This approach aims to reduce the need for chronic immunosuppressive treatments and frequent, invasive, and costly procedures [127]. Nevertheless, further research is needed to assess the feasibility of personalized therapy for conditions associated with dysbiosis. The application of engineered probiotics is limited by the current understanding of relevant biomarkers for gut inflammation and the number of characterized bacterial systems that can be reliably used [127]. Additionally, there is a risk of overproduction of the therapeutic substance at unwanted sites, potentially compromising both effectiveness and safety [127].

Probiotics are consumed in diverse ways, such as through dairy products, food supplements, and functional foods. Typically, most of the mentioned probiotic microorganisms can be found in different types of dairy and fermented milk products, like yogurt, cheese, or kefir [129,130]. However, emerging market trends are opening doors to new types of products, including those based on traditional methods of preserving plant-based products, like fermentation with lactic acid bacteria [131]. Among the most common plant-based products containing probiotics are olives, pickles, soy, coffee, sauerkraut, ogi, and kimchi [132,133,134]. Additionally, certain unfiltered fermented alcoholic beverages that still retain some microorganisms, such as beer, wine, and kombucha, as well as bakery products primarily using yeasts, can serve as sources of probiotics [135,136]. Consuming probiotics in this manner can facilitate their intake and may also promote synergies between different microbial genera [137]. This is noteworthy because, typically, various species or genera can perform similar beneficial functions for our body. However, consuming them in combination may offer additional advantages, as highlighted by Esposito et al., who demonstrated that the combination of *S. thermophilus* with several species of *Lactobacillus* and *Bifidobacteria*, can limit oxidative and inflammatory damage in nonalcoholic fatty liver disease [137].

## 3. Gut Dysbiosis and How It Is Affected by Diet

Gut dysbiosis is characterized by an imbalance in the composition of the gut microbiota, specifically regarding the relative abundances of different bacteria. This imbalance can be linked to functional alterations in the microbial transcriptome, proteome, or metabolome [138]. Notably, disruption in the Bacteroidota/Bacillota (Bacteroidetes/Firmicutes) ratio with increases in Enterobacterial populations, such as *Escherichia coli*, *Klebsiella* spp., and *Proteus* spp., are often seen in cases of gut dysbiosis [138].

A well-balanced gut microbiota is vital for maintaining intestinal stability and promoting human health. A wide range of both gastrointestinal and systemic conditions are linked to dysbiosis, such as IBD, obesity, diabetes, food allergies, asthma, and colorectal cancer, among others [18,19,139,140]. Increasing evidence indicates that dysbiosis in the gut microbiota and its metabolites can compromise the integrity of the intestinal barrier [141]. This disruption occurs by inhibiting the expression of proteins that are crucial for maintaining intestinal tight junctions [17], resulting in a greater passage of lipopolysaccharides (LPS) from the intestine into the bloodstream, which in turn leads to metabolic endotoxemia [142].

Gut dysbiosis can result from changes in diet, immune deficiencies, infections, or exposure to antibiotics and toxins [143]. In particular, diet can be considered as the main factor influencing gut microbiota throughout an individual’s life. In fact, nutrition is quite important from the earliest stages of life. Human breast milk, which is rich in oligosaccharides, supports the growth of bacteria that can process carbohydrates, like *Bifidobacterium* and *Bacteroides* species [144,145]. This leads to a distinct gut microbiota profile in breast-fed infants, while formula-fed infants tend to have higher levels of *Clostridium* spp. [144,145].

In adulthood, different types of diets can potentially influence the relative abundance of bacteria in the gut. On the one hand, Western diets that are high in fats and carbohydrates can lead to severe dysbiosis [146], decreasing the Bacteroidota/Bacillota ratio [146,147]. In a study conducted on mice, the gut microbiota of the low-fat diet group consisted of 61% Bacillota and 32% Bacteroidota, while the high-fat diet group showed a composition of 73% Bacillota and 21% Bacteroidota [148]. This shift has been associated with increased intestinal permeability and, consequently, with different metabolic disorders, such as obesity and type 2 diabetes, among others [146,149].

On the other hand, Mediterranean and vegetarian diets, with a lot of fruits, vegetables, olive oil, and oily fish, are known for their anti-inflammatory properties and might help prevent dysbiosis and the development of IBD [16,17]. In these types of diets, in addition to a greater Bacteroidota/Bacillota ratio, there are higher levels of SCFA producers and a slight decrease in intestinal pH, preventing the growth of potential pathogenic Enterobacteria, such as *E. coli* and others [150]. In addition, the intestinal microbiota has been reported to change depending on the type of fatty acid ingested. Omega-3 polyunsaturated fatty acids intake, characteristic of a Mediterranean diet, was directly associated with an increase in *Lactobacillus* abundance, while monounsaturated and omega-6 polyunsaturated fatty acids were related to decreased *Bifidobacterium* [151].

## 4. Intestinal Diseases and Relation with Probiotics

IBDs, such as Crohn’s disease or ulcerative colitis, are complex conditions with multiple contributing factors. Around 3.7 million individuals in Europe and the United States of America are affected by IBD [152]. These chronic, progressive immune disorders are associated with changes in microbiota composition, or dysbiosis, and an impaired mucosal barrier, which lead to excessive immunologic responses at the mucosal level [16]. Particularly, IBD is characterized by chronic inflammation, along with a concomitant production of elevated levels of pro-inflammatory cytokines and free radicals such as nitric oxide, which are likely involved in intestinal tissue injury [50,153]. Under normal physiological conditions, nitric oxide plays a role in the intestinal antibacterial response; for example, enteroinvasive bacteria like *E. coli*, *Salmonella*, and *Shigella* can directly induce inducible nitric oxide synthase (iNOS) expression as part of the host defense mechanism [154]. However, nitric oxide may occasionally become part of a dysregulated immune response, resulting in chronic inflammatory disorders such as IBD [155].

Studies with humans have determined different microbiota composition between IBD patients and healthy ones [13]. SCFA like propionic acid exhibits anti-inflammatory properties, and a reduction in SCFA-producing *Phascolarctobacterium* has been observed in IBD, exacerbating its symptoms [13]. Moreover, individuals with Chron’s disease or ulcerative colitis reported a reduction in anti-inflammatory bacteria, such as *Faecalibacterium prausnitzii*, in their fecal microbiota compared to healthy subjects [156,157]. However, it remains uncertain whether this reduction is a cause or a consequence of IBD.

Different probiotics have been tested to treat chronic diseases, mainly due to their ability to modulate the immune system and elicit an anti-inflammatory response by downregulating the production of inflammatory cytokines. Mice with dextran sulfate sodium (DSS)-induced colitis were treated with *L. acidophilus*, *B. lactis*, *L. plantarum*, and *B. breve* for 7 days [50]. This treatment improved clinical symptoms, histological alterations, and mucus production. In addition, probiotic supplementation decreased nitric oxide production by peritoneal macrophages compared to healthy mice [50]. Regarding human studies, patients with ulcerative colitis received twice daily for 12 weeks a probiotic preparation of four strains of *Lactobacillus* (*L. paracasei*, *L. plantarum*, *L. acidophilus*, and *L. delbrueckii* subspecies *bulgaricus*), three strains of *Bifidobacterium* (*B. longum*, *B. breve*, and *B. infantis*), and one strain of *S. thermophilus*, achieving a higher remission rate than control group [158]. However, the treatment of IBD with probiotics in humans remains controversial, as various clinical studies have failed to demonstrate significant improvements in these patients. For instance, Probio-Tec AB-25 (*L. acidophilus* La-5 and *B. animalis* subsp. *lactis* BB-12) showed no advantage over placebo in maintaining remission in patients with left-sided ulcerative colitis [159], and *B. breve* strain Yakult had no effect on time to relapse in ulcerative colitis patients [160]. This is not surprising, considering that IBD is a multifactorial disease and the microbiota is potentially influenced by diet, antibiotics, and other factors.

Another condition in which there is an imbalance in the gut microbiota is small intestinal bacterial overgrowth (SIBO). This is common in patients with irritable bowel syndrome and its diagnosis requires a hydrogen breath test, which detects hydrogen released from the fermentation of carbohydrates by gut bacteria [161]. The usual choice for managing SIBO is antibiotics; nevertheless, it is not always effective, and patients relapse after treatment [161]. Probiotics are emerging as a new approach to treat SIBO together with antibiotics; in fact, a meta-analysis has reported that probiotics improve SIBO by increasing the decontamination rate, reducing hydrogen concentration, and alleviating abdominal pain [162]. The probiotics studied included strains of *Lactobacillus* (such as *L. casei*, *L. acidophilus*, *L. rhamnosus*), *Bifidobacterium* (*B. breve*, *B. longum*, *B. infantis*), and *Bacillus clausii*, among others, either alone or in combination with antibiotics. However, probiotics have not been effective in preventing SIBO [162].

## 5. Dysbiosis and Probiotics in Metabolic Disorders

Obesity is a major public health issue today, influenced by a variety of factors and often associated with insulin resistance, which can lead to type 2 diabetes. A common underlying cause of obesity is an imbalance between energy intake and energy expenditure [19]. Emerging evidence indicates that this imbalance might be linked to an altered gut microbiota [163], as the gut’s bacterial communities play crucial roles in processes like digestion, nutrient absorption, and energy regulation [19]. In fact, studies have shown that, compared to lean mice, the gut microbiota in obese mice tends to have a lower proportion of Bacteroidota and a higher proportion of Bacillota [163].

Dysbiosis associated with obesity is closely tied to a high-fat diet and is characterized by a shift in the abundance of specific bacterial species and increased gut permeability [164,165]. Heightened intestinal permeability allows LPS to enter the bloodstream, leading to elevated levels that cause metabolic endotoxemia. This phenomenon is commonly observed in individuals with obesity, insulin resistance, or type 2 diabetes [166,167,168]. LPS are known for their pro-inflammatory properties, as they activate Toll-like receptors 4 (TLR4), nucleotide-binding oligomerization domain expression, and inflammasomes, which in turn promote the maturation of pro-inflammatory cytokines [19]. Consequently, chronic low-grade inflammation, changes in SCFA metabolism, and other factors like genetic predisposition, lifestyle, and diet, can drive the progression of metabolic disorders, such as obesity and insulin resistance.

The gut microbiota in obesity is characterized by the presence of genes that enhance energy harvest and metabolism, particularly those that encode enzymes for breaking down complex plant polysaccharides into SCFA [19]. These SCFA serve both as an energy source and as signaling molecules, influencing processes such as lipogenesis, fat storage, fatty acid oxidation, and gluconeogenesis [169]. SCFA, along with other microbial metabolites, are part of a balanced microbiota and promote gut health. Nevertheless, in the context of obesity, an imbalance in the microbiota can lead to elevated SCFA levels in the bloodstream, providing an extra energy source that may promote de novo lipogenesis in the liver [19]. The question remains whether higher SCFA levels in obese individuals are a cause or a consequence of such a condition.

Interventions using prebiotics and probiotics together can work synergistically to restore microbiota balance, reduce inflammation, and improve insulin resistance. Several clinical trials have shown that *Lactobacillus* and *Bifidobacterium* strains may help prevent metabolic disorders, including obesity [170,171]. A systematic review of 16,676 overweight and obese adults found that probiotics had a moderate effect on reducing body weight; however, these beneficial effects were only observed when probiotics were used in high doses [172]. In fact, the authors of the mentioned review highlight that most studies used probiotic mixtures, making it hard to identify the most effective strains. Variations in intervention duration, dosage, and participant characteristics could explain the discrepancies, emphasizing the need for further research on anti-obesity strains [172]. In addition, patients with type 2 diabetes were randomized to receive either 300 g of probiotic yogurt containing *L. acidophilus* La5 and *B. lactis* Bb12 or 300 g of conventional yogurt for 6 weeks. Probiotic consumption led to significant decreases in total cholesterol, low-density lipoprotein cholesterol (LDL-C), and atherogenic indices compared to controls, suggesting an improvement in cardiovascular disease risk factors [173].

Prebiotics like oligofructose, long-chain inulin, or β-glucans have also demonstrated not only improvements in gut microbiota but beneficial effects on metabolic disorders. A randomized controlled trial found that supplementing with oligofructose-enriched inulin helped manage pediatric overweight and obesity by improving appetite control and reducing energy intake in children aged 11–12 [174]. The authors emphasize the need for further research to clarify the mechanisms behind these physiological effects. One proposed explanation is enhanced satiety, as SCFA bind to specific receptors on colonic L-cells (free fatty acid receptors or FFAR), stimulating the release of appetite-regulating hormones [175]. One proposed explanation is enhanced satiety, as SCFA binds specific receptors on colonic L-cells (free fatty acid receptors or FFAR), stimulating the release of appetite-regulating hormones [175]. In addition, type 2 diabetic women who received a daily dose of 10 g of oligofructose-enriched inulin showed significant improvements in glycemic status, lipid profile, and immune markers [176].

## 6. Gut Microbiota and Probiotics in Neurological Diseases

The knowledge of the relationship between the intestine, gut microbiota, and brain diseases has rapidly increased in the last 15 years, with the gut microbiota sometimes being a key factor in the susceptibility and development of certain pathologies such as Alzheimer’s disease (AD), Parkinson’s disease (PD), autism, and multiple sclerosis [21,177]. The gut microbiota exhibits an important place in the crosstalk between the gut and the brain though the vagus nerve. Several theories have been proposed to explain the communication pathways between the gut and the brain, including the neuroendocrine hypothalamic–pituitary–adrenal axis, the neuroanatomical gut–brain axis, the gut immune system, the gut microbiota metabolism system, the intestinal mucosal barrier, and the blood–brain barrier [178,179]. The gut microbiota participates in this communication by synthesizing neurotransmitters such as gamma-aminobutyric acid (GABA), catecholamines, serotonin, acetylcholine, and dopamine, which may modify host neuronal activity. Additionally, it produces SCFAs, primarily acetate, propionate, and butyrate [180,181]. Moreover, it can reduce cortisol levels, lipid peroxidation, and monoamine oxidase activity or modulate specific minerals in tissues, such as magnesium and zinc [182,183,184].

The connection between dysbiosis and inflammation it generates has already been discussed. This inflammation can lead to an acceleration of certain diseases, where a pro-neuroinflammatory environment worsens disease progression [185]. However, some diseases go beyond merely worsening prognoses, being directly associated with specific dysbiosis [186]. On the one hand, numerous clinical studies have found that dysbiosis in the small intestine can influence the progression of PD, by increasing neuroinflammation and α-synuclein aggregation, or by decreasing SCFA levels [15]. These SCFA are activators of G protein-coupled receptors, inhibiting histone deacetylases and leading to epigenetic regulation of antioxidant genes and redox signaling. Also, some SCFA have shown protective activity against dopaminergic neuron loss, while inhibiting neuroinflammation and the motor dysfunction characteristic of PD [187,188,189]. On the other hand, increases in pathogenic bacteria such as *Escherichia* or *Shigella* have been linked to elevated levels of pro-inflammatory cytokines such as C-X-C motif chemokine ligand 2 (CXCL2), interleukins (IL-1β, IL-6), or inflammasome complexes (NLR family pyrin domain containing 3 or NLRP3) in patients with brain amyloidosis—an accumulation of amyloid proteins in the brain—worsening the pathogenesis of AD [190]. Therefore, these alterations in the gut microbiome may act synergistically with genetic or environmental factors to increase the risk of developing a neurological disorder.

Similarly, just as an imbalance in the gut microbiota can lead to or accelerate the progression of a neurological disease, the use of probiotics can be employed to regulate certain neuronal pathologies. For example, *L. plantarum* can reduce α-synuclein accumulation in the substantia nigra in PD, while *B. animalis* and *L. acidophilus* have been shown to rescue nigral dopaminergic neurons from 1-methyl-4-phenyl-1,2,3,6-tetrahydropyridine (MPTP) and rotenone-induced neurotoxicity [191]. Generally, these approaches involve administering probiotics or fecal microbiota transplantation to the patient, which can lead to SCFA production in the intestine, helping to reduce intestinal inflammation in PD and α-synuclein aggregation [192]. Two recent meta-analysis studies showed how oral probiotic consumption, mainly *Lactobacillus* and *Bifidobacterium* of different species, significantly improved motor symptoms, gastrointestinal dysfunction, anxiety, and depression in patients with PD [193,194,195]. A similar trend is observed in studies related to AD, where probiotics have been shown to improve cognitive decline, although this effect appears to be more pronounced in individuals with mild cognitive impairment, which is considered a prodromal stage of clinical AD [195,196]. One aspect to consider is that these kinds of studies apply preliminary filtering, which may exclude inconclusive trials, and include studies in which the probiotics used are not the same or consist of various genera and species, despite reaching a uniform conclusion.

Moreover, new applications for probiotics are emerging, providing mental health benefits by inducing metabolites, hormones, and immune factors, and by exhibiting antidepressant and anxiolytic activity, thus falling under the category of psychobiotics [197]. For example, *B. breve* has improved cognitive function in older adults with suspected mild cognitive impairment [93], *L. rhamnosus* has shown anxiolytic capacity [86], *L. casei* has decreased anxiety [62], and *B. infantis* has normalized behavior [96]. However, a recent meta-analysis revealed that only a limited number of probiotic species are associated with a reduction in anxiety in animals. This was observed, for instance, with *L. rhamnosus*. Nevertheless, in human studies, this strain did not produce significant improvements in anxiety symptoms [198].

It is well established that probiotics can mitigate the dysbiosis induced by antibiotics, which may result in neural alterations, and support the recovery of patients who have undergone neurosurgical procedures. In mice, treatment with ampicillin has been shown to reduce intestinal populations of *Lactobacillus*, *Bifidobacterium*, *Clostridium*, and Bacillota, resulting in decreased intestinal crypt depth and villous length, as well as impairments in cognition and hippocampal neuronal density. Furthermore, antibiotic treatment also increased corticohippocampal acetylcholinesterase activity, myeloperoxidase activity, and oxidative stress. These changes were partially reversed by treatment with the probiotic *Bifilac*, which contains *Lactobacillus sporogenes*, *Clostridium butyricum*, *Streptococcus faecalis*, and *Bacillus mesentericus* [199].

Finally, probiotics have shown some benefits in neurosurgical and trauma patients by improving gastrointestinal function, modulating inflammation, and reducing infection rates [200]. A multi-strain probiotic including *B. animalis*, *B. bifidum*, *B. longum*, *L. rhamnosus*, *L. plantarum*, *L. acidophilus*, and *S. thermophilus* significantly improved gastrointestinal motility after brain tumor surgery [201]. In patients with traumatic brain injury, studies using combinations of *B. longum*, *L. acidophilus*, *L. bulgaricus*, *E. faecalis*, and *S. thermophilus* showed reduced inflammatory cytokines such as IL-6, interferon-gamma (IFN-γ), or tumor necrosis factor-alpha (TNF-α), alongside improved immune responses, enhancing immune functioning and decreasing the incidence rate of complications [202,203,204]. One of the main limitations associated with this type of clinical trial is the typically small sample size of the study population. Although *L. johnsonii* demonstrated reduced intensive care unit stays, fewer infections, and overall improved recovery in brain injury patients, it is important to note that the clinical trial included only 10 individuals in the probiotic group and 10 in the control group [205]. Similarly, in a study involving *E. faecium* in patients after spinal surgery, the probiotic group (*n* = 16) showed a decrease in the abundance of the harmful gut bacteria *Streptococcus gallolyticus* compared to the control group (*n* = 17), with no significant differences in alpha nor beta diversity of gut microbiota, suggesting protection against pathogenic bacteria without significant changes in general microbiota [206]. Finally, a probiotic mix including *L. acidophilus*, *B. animalis*, *L. plantarum*, and *S. boulardii* reduced surgical site infections after trauma surgeries (103 patients), including 19 patients in the neurosurgery subgroup, 13 in the control group, and 6 in the probiotic group [207]. Therefore, due to the limited sample sizes often found in these studies, the observed results should be interpreted with caution.

## 7. Dysbiosis and Probiotics in the Immune System

In this review, we observed how the intestinal microbiota, and, consequently, probiotics, are associated with various intestinal, metabolic, and neurological diseases. Many of these diseases are interconnected through the immune system or by the disruption of the gut barrier due to dysbiosis [208]. Moreover, intestinal dysbiosis has been implicated as a potential trigger for immune-mediated diseases such as rheumatoid arthritis [209], systemic lupus erythematosus, multiple sclerosis (MS) [210], and IBD [16].

Microbial–immune crosstalk involves SCFA signaling, tryptophan metabolism via aryl hydrocarbon receptors, nucleoside signaling, and activation of intestinal histamine-2 receptors [211]. On the one hand, gut microbiota can influence innate immunity. As mentioned earlier, increased intestinal permeability can lead to elevated levels of LPS in the bloodstream, triggering a pro-inflammatory state. LPS activate TLR4, leading to the activation of nuclear factor kappa B (NF-κB) signaling and the production of pro-inflammatory cytokines (TNF-α, IL-1β, IL-6) [19,212]. Additionally, other microbial components can stimulate nucleotide-binding oligomerization domain proteins and inflammasomes, further promoting inflammatory responses [19]. On the other hand, the intestinal microbiota can also impact adaptive immunity by influencing secretory IgA levels. Certain bacterial species have been associated with lower secretory IgA levels [213], which may compromise mucosal immunity and alter host-microbiota interaction [214]. However, rather than direct degradation of IgA by the microbiota, this effect may be due to broader immune system dysregulation.

Probiotics like *B. animalis*, *L. acidophilus*, *L casei*, *L. johnsonii*, or *L. rhamnosus* can enhance innate and nonspecific cellular immune responses thought the activation of macrophages, dendritic cells, TLR, natural killer cells, and B or T lymphocytes [20]. On the one hand, in rats, *L. acidophilus* has been reported to inhibit the expression of the Niemann-Pick C1-like 1 (*NPC1L1*) gene in the small intestine and regulate levels of oxidized LDL, superoxide dismutase (SOD), TNF-α, and IL-10, thereby suppressing inflammation and oxidative stress, and inhibiting the development of atherosclerosis effects [42]. Furthermore, *L. acidophilus* strains ATCC 314 and PTCC 1643 have exerted anti-inflammatory properties in an arthritis rat model and modulated the expression of TLR2 and TLR4 in HT29 intestinal epithelial cells [42]. On the other hand, in mice, oral administration of *L. casei* induced an early innate immune response, increasing CD206, a receptor of macrophages and dendritic cells, and TLR2 markers [215].

In humans, different species and strains of *Lactobacillus* and *Bifidobacterium* have shown an increase in anti-*Salmonella typhi* antibody response and serum IgA levels [216,217], an increase in serum IgG during early response (0–21 days), and an increase in IgA and IgM in late response (21–28 days) [218]. Randomized controlled trials suggest that probiotic-induced microbial modulation may alleviate gastrointestinal symptoms and systemic inflammation in conditions such as rheumatoid arthritis, ulcerative colitis, and MS [211]. In patients with rheumatoid arthritis, the supplementation with *L. casei* was associated with favorable changes in circulating IL-10, IL-12, and TNF-α levels [219]. *L. reuteri* reduced pro-inflammatory cytokines such as TNF-α, IL-1, and IL-8, proposing an effective probiotic treatment against distal ulcerative colitis [220]. Respect MS, two studies showed a reduction in abundance of *Prevotella* and *Lactobacilli*, among others, compared to healthy controls, pointing a possible alteration of gut microbiota [210,221], while the probiotic *L. reuteri* suggested some improvements in inflammatory factors, mental health, and expanded disability status scale scoring [222]. Furthermore, a recent meta-analysis highlighted probiotic supplementation as a therapeutic option, improving depression symptoms and inflammatory status in patients with MS [223]. All these factors suggest that alterations in the gut microbiota may lead to immunological changes in the host, potentially triggering or exacerbating certain immune-related diseases. Conversely, restoring the microbiota or introducing beneficial microorganisms that help reestablish immunological balance may alleviate symptoms and improve the prognosis of these conditions.

## 8. Conclusions

Current knowledge about the human microbiome and the role of probiotics in disease prevention and treatment is rapidly expanding. The gut microbiota and its complex interaction with human health represent a dynamic field of research, further complicated by individual variability and strain-specific effects. Probiotics have gained growing interest for their ability to support a balanced gut microbiota, which is vital for digestion, immunity, and mental health. As modern diets and lifestyles increase the risk of dysbiosis, interest in probiotic-based strategies continues to rise.

This review explores the interplay between multiple pathologies, their associated gut microbiota imbalances, and the prospective use of probiotics as a therapeutic strategy. In particular, we highlight common features across the diseases examined, including immune involvement, chronic inflammation, and the emergence of pathogenic microorganisms. Nonetheless, we also underscore the multifactorial nature of these disorders and the inherent challenges in disentangling their biological complexity.

Although many questions remain unanswered, the future of probiotics is promising. Continued research may lead to personalized approaches tailored to individual microbiota profiles, offering new tools for the prevention and management of complex diseases. To fully realize their therapeutic potential, improvements in clinical trial design are essential. These include the use of well-characterized, isolated strains rather than probiotic mixtures, increased sample sizes to ensure statistical power, and longer intervention periods to better assess sustained effects.

## Figures and Tables

**Figure 1 microorganisms-13-01084-f001:**
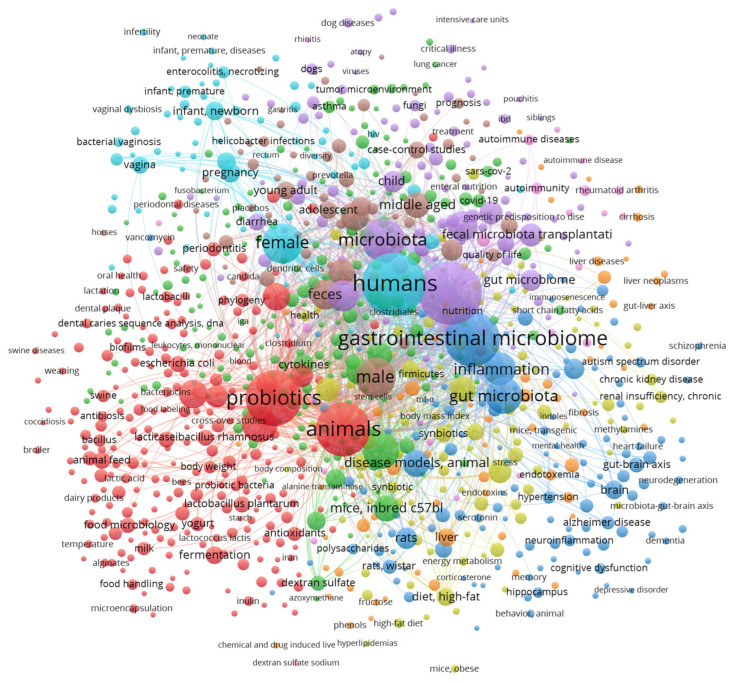
Detailed VOSViewer bibliographic analysis for the keywords “Probiotic” and “Gut microbiota”, database source: https://pubmed.ncbi.nlm.nih.gov/ (accessed on 3 December 2024). Colors of nodes represent different clusters.

**Table 1 microorganisms-13-01084-t001:** Most relevant probiotic microorganisms with their action mechanisms and health benefits. New genera are included between parentheses according to the LPSN—List of Prokaryotic names with Standing in Nomenclature.

Genera	Species	Mechanisms of Action/Benefits	References
* Lactobacillus *	* acidophilus *	1, 2, 3, 4, 5, 6, 7, 8, 9, 10, 11, 13, 15, 16, 17	[25,40,42,43,44,45,46,47,48,49,50,51]
	* amylovorus *	1, 2, 4, 6, 10	[52]
(*Levilactobacillus*)	* brevis *	1, 3, 4, 6, 7, 9, 11, 13, 14, 17	[48,53,54,55,56,57,58]
(*Lacticaseibacillus*)	*casei*	1, 3, 4, 6, 7, 10, 11, 12, 14, 16, 17	[40,48,49,59,60,61,62,63]
	* crispatus *	1, 4, 13, 14	[64,65,66]
	* delbrueckii * ssp. *bulgarius*	1, 3, 4, 6, 7, 10, 16	[44,48,49,67]
	* delbrueckii * ssp. *lactis*	1, 2, 4, 7, 11	[40,68]
(*Limosilactobacillus*)	* fermentum *	1, 2, 4, 5, 6, 7, 8, 10, 13	[48,49,51,57,69]
	* gasseri *	1, 2, 4, 8, 11, 13, 14, 15	[40,48,51,66,70]
	* helveticus *	1, 4, 5, 6, 7, 12, 13, 17	[40,47,60,71]
	* iners **	1, 13	[66,72]
	* jensenii *	1, 4, 6, 7, 13	[66,73]
	* johnsonii *	1, 4, 6, 7, 10	[44,48,74,75]
	* kefiranofaciens *	1, 4, 6, 7, 11	[76]
(*Lacticaseibacillus*)	* paracasei *	1, 4, 6, 7, 14, 15, 16	[45,48,63,77,78,79]
(*Lactiplantibacillus*)	* pentosus *	1, 2, 3, 4, 7, 8, 16	[48,80,81,82]
(*Lactiplantibacillus*)	* plantarum *	1, 2, 3, 4, 6, 7, 8, 10, 11, 13, 14, 16	[25,48,49,50,51,57,78,80]
(*Limosilactobacillus*)	* reuteri *	1, 6, 9, 10, 13, 14, 15, 17	[25,47,48,63,83,84,85]
(*Lacticaseibacillus*)	* rhamnosus *	1, 3, 4, 7, 8, 10, 11, 12, 13, 14, 15, 17	[25,40,45,48,49,60,63,71,79,86]
(*Ligilactobacillus*)	* salivarius * ssp. *salicinius*	1, 2, 9, 13, 14, 17	[25,40,47,51,63]
* Bifidobacterium *	* adolescentis *	1, 2, 4, 7, 11, 17	[47,51,87,88,89]
	* animalis *	1, 3, 4, 6, 11, 14, 16	[35,49,71,79]
	* animalis * ssp. *lactis*	1, 3, 4, 6, 7, 10, 11, 14, 15, 16, 17	[45,46,47,48,50,63,79,90,91]
	* breve *	1, 4, 6, 7, 10, 11, 17	[40,47,48,49,50,71,87,92,93]
	* bifidum *	1, 2, 3, 4, 7, 10, 11, 14, 15, 17	[40,46,47,48,51,87,94]
	* dentium *	1, 4, 7, 14	[95]
	*longum* spp. *infantis*	1, 3, 4, 6, 7, 10, 11, 17	[40,47,48,89,96]
	* longum *	1, 2, 3, 4, 6, 7, 10, 11, 17	[40,47,48,49,51,61,67,71,87,89]
	* pseudocatenulatum *	1, 2, 4, 5, 11	[40,97,98,99,100,101]
	* thermophilum *	1, 11, 14	[40,102]
* Enterococcus *	* durans ***	1, 8	[103]
	* faecalis *	1, 3, 4, 10	[48,104,105]
	* faecium *	1, 9, 10	[25,106]
* Lactococcus *	* lactis * ssp. *cremoris* **	1, 2, 4, 8, 16	[71,107,108,109]
	* lactis * ssp. * lactis ***	1, 4, 8, 17	[47,109,110,111]
	* lactis * ssp. * lactis bv. diacetylactis ***	1, 2, 8, 9	[109,110,111]
* Streptococcus *	* salivarius *	1, 14	[112,113]
	* thermophilus ***	1, 4, 6, 7, 9, 10	[48,49,50,71]
* Propionibacterium *	* freudenreichii *	1, 2, 3, 4, 6, 7	[38,48,51,71,114]
(*Acidipropionibacterium*)	* acidipropionici ***	1, 2, 7	[38,89]
(*Acidipropionibacterium*)	* jensenii ***	1, 2, 7	[38,89]
(*Acidipropionibacterium*)	* thoenii ***	1, 2, 7	[38,89]
* Leuconostoc *	* mesenteroides * ssp. *cremoris* *	1, 2, 4, 14	[51,71,115,116]
* Pediococcus *	* acidilactici *	1, 4, 13, 17	[48,117]
	* pentosaceus *	1, 3, 4, 8, 9	[118]

Adapted from [25,40,47,48,119]. Mechanisms of action/benefits: 1 (decrease intestinal pH and regulate microflora/prevention of infections), 2 (produce metabolites/vitamins/enzymes that improve nutrition), 3 (inhibit carcinogenic enzymes, anticarcinogenic properties), 4 (modulate immunity through anti-inflammatory or pro-inflammatory cytokines), 5 (reduce cardiovascular diseases), 6 (beneficial role against intestinal diseases like inflammatory intestinal injury, ulcerative colitis, decline crypt depth), 7 (reduce oxidative stress/present antioxidant activity), 8 (control cholesterol or lipid levels in blood), 9 (specific antibiotics/bacteriocins production), 10 (help against diarrhea), 11 (positive effect against metabolic diseases like obesity or diabetes), 12 (improve lung or kidney pathologies), 13 (prevention of urogenital infections/positive in reproductive health), 14 (buccal health improvement), 15 (skin diseases improvement/atopic dermatitis treatment), 16 (reduce respiratory tract infections), 17 (prevention/improvement of neural diseases or disorders). * Bacteria with a controversial role as probiotics. ** Promising possible probiotics.

## Data Availability

No new data were created or analyzed in this study. Data sharing is not applicable to this article.

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
