# Peer review of "Gut Microbiota: An Immersion in Dysbiosis, Associated Pathologies, and Probiotics"

_microorganisms, 2025, doi:10.3390/microorganisms13051084_

Round 1

Reviewer 1 Report

Comments and Suggestions for Authors

Dear authors, I found your manuscript not accetable for pubblication because it looks like a replication of several reviews already published.

Comments on the Quality of English Language

The quality of English language needs an improvement.

Author Response

Comment 1: Dear authors, I found your manuscript not accetable for pubblication because it looks like a replication of several reviews already published.

Response 1: 

We thank Reviewer 1 for having taken the time to evaluate our manuscript. While we acknowledge the reviewer’s recommendation, no specific comments or suggestions for improvement were provided. Nevertheless, we have thoroughly revised the manuscript in response to the constructive feedback received from the other reviewers, and we believe these changes have significantly strengthened the quality and clarity of our work. We respectfully submit the revised version for further consideration.

Reviewer 2 Report

Comments and Suggestions for Authors

Major:

  1. The authors should provide a more balanced discussion that includes studies with conflicting or inconclusive results regarding probiotic effectiveness.
  2. The authors should clearly distinguish between correlation and causation when discussing microbiota composition and disease associations. Statements implying causality should be supported by mechanistic or interventional evidence.
  3. The authors should expand on the engineered probiotics section by including more details on their mechanisms, clinical applicability, and current limitations.

Minor:

  1. The authors should ensure consistent and correct usage of terms such as "microbiota" and "microbiome", as these are occasionally used interchangeably.
  2. The authors should ensure all references follow the journal's required citation format, including uniform use of punctuation and DOIs.
Comments on the Quality of English Language

Minor linguistic and stylistic corrections are required.

Author Response

Major:

  1. The authors should provide a more balanced discussion that includes studies with conflicting or inconclusive results regarding probiotic effectiveness.

Thank you for your comment. We agree that including studies with conflicting or inconclusive results is essential for a balanced discussion.

We have expanded the controversial effects of probiotics in IBD: page 7, lines 254-265 “Regarding human studies, patients with ulcerative colitis received twice daily for 12 weeks a probiotic preparation of 4 strains of Lactobacillus (L. paracasei, L. plantarum, L. acidophilus, and L. delbrueckii subspecies bulgaricus), 3 strains of Bifidobacterium (B. longum, B. breve, and B. infantis), and 1 strain of S. thermophilus, achieving a higher remission rate than control group. However, the treatment of IBD with probiotics in humans remains controversial, as various clinical studies have failed to demonstrate significant improvements in these patients. For instance, Probio-Tec AB-25 (L. acidophilus La-5 and B. animalis subsp. lactis BB-12) showed no advantage over placebo in maintaining remission in patients with left-sided ulcerative colitis, and B. breve strain Yakult had no effect on time to relapse in ulcerative colitis patients. This is not surprising, considering that IBD is a multifactorial disease and the microbiota is potentially influenced by diet, antibiotics, and other factors.”.

Additionally, in the mentioned section we already had this phrase regarding SIBO: page 8, lines 270-277 “Probiotics are emerging as a new approach to treat SIBO together with antibiotics; in fact, a meta-analysis has reported that probiotics improve SIBO by increasing the decontamination rate, reducing hydrogen concentration, and alleviating abdominal pain. The probiotics studied included strains of Lactobacillus (such as L. casei, L. acidophilus, L. rhamnosus), Bifidobacterium (B. breve, B. longum, B. infantis), and Bacillus clausii, among others, either alone or in combination with antibiotics. However, probiotics were not effective in preventing SIBO.”.

In the section of metabolic disorders, we had already written: page 8, lines 311-314 “A systematic review of 16676 overweight and obese adults found that probiotics had a moderate effect on reducing body weight; however, these beneficial effects were only observed when probiotics were used in high doses”. Nevertheless, to provide a more detailed discussion, we have added the following lines: page 8, lines 314-317 “In fact, the authors of the mentioned review highlight that most studies used probiotic mixtures, making it hard to identify the most effective strains. Variations in intervention duration, dosage, and participant characteristics could explain the discrepancies, emphasizing the need for further research on anti-obesity strains.”.

In the section of neurological diseases we had already written: page 10, lines 372- 375 “For example, L. plantarum can reduce α-synuclein accumulation in the substantia nigra in Parkinson’s disease, while B. animalis and L. acidophilus have been shown to rescue nigral dopaminergic neurons from 1-methyl-4-phenyl-1,2,3,6-tetrahydropyridine (MPTP) and rotenone-induced neurotoxicity.”.

Furthermore, we have added an additional paragraph discussing:

  • 3 recent meta-analysis about Parkinson’s disease, Alzheimer’s disease, and mild cognitive impairment where all of the studies showed positive effect on probiotic use: page 10, lines 378-387 “Two recent meta-analysis studies showed how oral probiotic consumption, mainly Lactobacillus and Bifidobacterium of different species, significantly improved motor symptoms, gastrointestinal dysfunction, anxiety, and depression in patients with PD. A similar trend is observed in studies related to AD, where probiotics have been shown to improve cognitive decline, although this effect appears to be more pronounced in individuals with mild cognitive impairment, which is considered a prodromal stage of clinical AD. One aspect to consider is that these kinds of studies apply preliminary filtering, which may exclude inconclusive trials, and include studies in which the probiotics used are not the same or consist of various genera and species, despite reaching a uniform conclusion.”.
  • 1 recent meta-analysis about anxiety and probiotics, where non-significant decrease of anxiety was found in humans: page 10, lines 393-397 “However, a recent meta-analysis revealed that only a limited number of probiotic species are associated with a reduction in anxiety in animals. This was observed, for instance, with L. rhamnosus. Nevertheless, in human studies, this strain did not produce significant improvements in anxiety symptoms.”.
  • Another application of probiotics in relation to neurobiology and postoperative contexts, in which we highlighted the limited sample sizes often found in these types of studies. Therefore, the observed results should be interpreted with caution: pages 10-11, lines 408-430 “Finally, probiotics have shown some benefits in neurosurgical and trauma patients by improving gastrointestinal function, modulating inflammation, and reducing infection rates. A multi-strain probiotic including B. animalis, B. bifidum, B. longum, L. rhamnosus, L. plantarum, L. acidophilus, and S. thermophilus significantly improved gastrointestinal motility after brain tumor surgery​. In patients with traumatic brain injury, studies using combinations of B. longum, L. acidophilus, L. bulgaricus, E. faecalis, and S. thermophilus showed reduced inflammatory cytokines such as IL-6, interferon-gamma (IFN-γ), or tumor necrosis factor-alpha (TNF-α), alongside improved immune responses, enhancing immune functioning and decreasing the incidence rate of complications. One of the main limitations associated with this type of clinical trial is the typically small sample size of the study population. Although L. johnsonii demonstrated reduced intensive care unit stays, fewer infections, and overall improved recovery in brain injury patients, it is important to note that the clinical trial included only 10 individuals in the probiotic group and 10 in the control group.​ Similarly, in a study involving E. faecium in patients after spinal surgery, the probiotic group (n = 16) showed a decrease in the abundance of the harmful gut bacteria Streptococcus gallolyticus compared to the control group (n = 17), with no significant differences in alpha nor beta diversity of gut microbiota, suggesting protection against pathogenic bacteria without significant changes in general microbiota. Finally, a probiotic mix including L. acidophilus, B. animalis, L. plantarum, and S. boulardii reduced surgical site infections after trauma surgeries (103 patients), including 19 patients in the neurosurgery subgroup, 13 in the control group, and 6 in the probiotic group. Therefore, due to the limited sample sizes often found in these studies, the observed results should be interpreted with caution.”.

Finally, in ‘Dysbiosis and probiotics in the immune system”, we have included additional information about randomized controlled trials in rheumatoid arthritis, ulcerative colitis, and multiple sclerosis: pages 11-12, lines 467-484 “Randomized controlled trials suggest that probiotic-induced microbial modulation may alleviate gastrointestinal symptoms and systemic inflammation in conditions such as rheumatoid arthritis, ulcerative colitis, and MS. In patients with rheumatoid arthritis, the supplementation with L. casei was associated with favorable changes in circulating IL-10, IL-12, and TNF-α levels. L. reuteri reduced pro-inflammatory cytokines such as TNF-α, IL-1, and IL-8, proposing an effective probiotic treatment against distal ulcerative colitis. Respect MS, two studies showed a reduction of abundance of Prevotella and Lactobacilli, among others, compared to healthy controls, pointing a possible alteration of gut microbiota, while the probiotic L. reuteri suggested some improvements in inflammatory factors, mental health, and expanded disability status scale scoring. Furthermore, a recent meta-analysis highlighted probiotic supplementation as a therapeutic option, improving depression symptoms and inflammatory status in patients with MS. All these factors suggest that alterations in the gut microbiota may lead to immunological changes in the host, potentially triggering or exacerbating certain immune-related diseases. Conversely, restoring the microbiota or introducing beneficial microorganisms that help reestablish immunological balance may alleviate symptoms and improve the prognosis of these conditions.”.

  1. The authors should clearly distinguish between correlation and causation when discussing microbiota composition and disease associations. Statements implying causality should be supported by mechanistic or interventional evidence.

Thank you for your comment. In some sections of our review, we had already stated that it is unclear whether intestinal dysbiosis is a cause or a consequence of the disease, as in the following examples:

  • ‘4. Intestinal diseases and relation with probiotics’: page 7, lines 246-247 “However, it remains uncertain whether this reduction is a cause or a consequence of IBD.”.
  • ‘5. Dysbiosis and probiotics in metabolic disorders’: page 8, lines 295-297 “Consequently, chronic low-grade inflammation, changes in SCFA metabolism, and other factors like genetic predisposition, lifestyle, and diet, can drive the progression of metabolic disorders, such as obesity and insulin resistance.” or page 8, lines 306-307 “The question remains whether higher SCFA levels in obese individuals are a cause or a consequence of such condition.”.

In the ‘6. Gut microbiota and probiotics in neurological diseases”, we had written that gut dysbiosis may promote the progression of Parkinson’s and Alzheimer’s diseases: page 9, lines 356-358 “On the one hand, numerous clinical studies have found that dysbiosis in the small intestine can influence the progression of Parkinson’s disease, by increasing neuroinflammation and α-synuclein aggregation, or by decreasing SCFA levels.”; page 9, lines 362-367 "On the other hand, increases in pathogenic bacteria such as Escherichia or Shigella have been linked to elevated levels of pro-inflammatory cytokines such as C-X-C motif chemokine ligand 2 (CXCL2), interleukins (IL-1β, IL-6), or inflammasome complexes (NLR family pyrin domain containing 3 or NLRP3) in patients with brain amyloidosis—an accumulation of amyloid proteins in the brain—worsening the pathogenesis of Alzheimer's disease.”.

However, we have added now the phrase: pages 9-10, lines 367-369 “Therefore, these alterations in the gut microbiome may act synergistically with genetic or environmental factors to increase the risk of developing a neurological disorder.”.

In the ‘7. Dysbiosis and probiotics in the immune system’ section is more difficult to define the correlation/causation. However, we have included information about known changes in gut microbiota in relation with different inflammatory diseases and we have hypothesized that: page 12, lines 479-481 ”All these factors suggest that an alteration in the gut microbiome may lead to immunological changes in the host, potentially triggering or exacerbating certain immune-related diseases.”.

  1. The authors should expand on the engineered probiotics section by including more details on their mechanisms, clinical applicability, and current limitations.

We have expanded the section on engineered probiotics to include more details on their mechanisms of action, potential clinical applications, and current limitations: page 5, lines 148-160 “Currently, there is ongoing research into the use of engineered probiotics that could act as ‘sense and response’ systems (biosensor and biotherapeutic). These engineered probiotics would be bacteria that respond to specific biomarkers of inflammation by producing a therapeutic molecule, typically through transfected plasmids encoding for immunoregulatory cytokines or anti-inflammatory mediators, which are activated only upon induction of specific promoters. This approach aims to reduce the need for chronic immunosuppressive treatments and frequent, invasive, and costly procedures. Nevertheless, further research is needed to assess the feasibility of personalized therapy for conditions associated with dysbiosis. The application of engineered probiotics is limited by the current understanding of relevant biomarkers for gut inflammation and the number of characterized bacterial systems that can be reliably used. Additionally, there is a risk of overproduction of the therapeutic substance at unwanted sites, potentially compromising both effectiveness and safety.”.

Additionally, we have moved this paragraph from the “Intestinal diseases and relation with probiotics” section to “Gut microbiota, probiotics, and mechanisms of action” to present engineered probiotics as a versatile tool potentially applicable to various conditions.

Minor:

  1. The authors should ensure consistent and correct usage of terms such as "microbiota" and "microbiome", as these are occasionally used interchangeably.

We apologize for this mistake. The terms "microbiota" and "microbiome" have been carefully reviewed throughout the manuscript to ensure their consistent and accurate usage. Necessary modifications have been made where inconsistencies were identified.

  1. The authors should ensure all references follow the journal's required citation format, including uniform use of punctuation and DOIs.

All references have been carefully reviewed to ensure they adhere to the required formatting guidelines.

Reviewer 3 Report

Comments and Suggestions for Authors

1. The manuscript extensively reviews well-known topics such as gut microbiota, dysbiosis, and probiotics without providing substantial new insights or novel perspectives.

2. Its overly broad scope leads to superficial coverage of multiple subjects, resulting in a lack of detailed analysis or depth.

3. The authors primarily summarize previously published studies without sufficiently critical evaluation or meaningful synthesis of findings.

4. Clinical applicability and practical implications remain uncertain due to contradictory findings and inconsistent evidence presented throughout.

5. Structural issues, including repetitive content and poor narrative organization, significantly compromise the manuscript's overall readability and coherence.

Author Response

  1. The manuscript extensively reviews well-known topics such as gut microbiota, dysbiosis, and probiotics without providing substantial new insights or novel perspectives.

Thank you for your comment. In response, we have incorporated additional information throughout the manuscript to provide a more in-depth analysis of the main controversies, critical limitations, and instances of insufficient efficacy associated with certain probiotic applications in each section. We believe this offers less commonly discussed perspectives of probiotics.

We have also included the updated taxonomy of all probiotic microorganisms (primarily reflected in the Table and in section 2), an aspect not included in recent reviews of this type. Additionally, we have explored in greater depth the mechanisms of action of each probiotic microorganism, highlighting why and how they may exert beneficial effects. To our knowledge, no previous review has covered such a wide range of probiotic species and benefits while incorporating up-to-date references. Finally, we have addressed emerging and lesser-known topics such as psychobiotics, engineered probiotics, and novel applications of probiotics in the treatment of autoimmune diseases like multiple sclerosis, supported by several meta-analyses. We hope these additions strengthen the novelty and relevance of our review.

  1. Its overly broad scope leads to superficial coverage of multiple subjects, resulting in a lack of detailed analysis or depth.

Thank you for your observation. We acknowledge that the broad scope of this review may inherently limit the level of detail dedicated to each individual topic. However, this was an intentional choice aimed at providing a comprehensive and integrative overview of the role of probiotics in the context of various health conditions associated with dysbiosis. Given the growing recognition of the microbiota’s involvement in a wide array of diseases, we believe that such a wide-angle perspective is valuable to contextualize the therapeutic potential of probiotics within the broader framework of human health.

Nonetheless, we would like to emphasize that significant efforts were made to deepen the analysis in several key areas. Specifically, we have expanded the review by incorporating numerous recent clinical trials, explicitly specifying the probiotic strains used across all sections (intestinal, metabolic, neuronal, and immune-related conditions). Additionally, we have provided a more detailed discussion on selected autoimmune diseases, such as multiple sclerosis, and included emerging topics such as engineered probiotics and psychobiotics—subjects that are often underrepresented in previous reviews. These additions aim to strike a balance between breadth and depth, and to enhance the informative value of the manuscript for both researchers and clinicians.

  1. The authors primarily summarize previously published studies without sufficiently critical evaluation or meaningful synthesis of findings.

We appreciate this important observation. In response, we have carefully revised the manuscript to include a more critical evaluation of the studies discussed throughout each section. We have emphasized differences in study design, population characteristics, strain specificity, and outcomes, highlighting both consistencies and discrepancies across the literature.

We have expanded the controversial effects of probiotics in IBD: page 7, lines 254-265 “Regarding human studies, patients with ulcerative colitis received twice daily for 12 weeks a probiotic preparation of 4 strains of Lactobacillus (L. paracasei, L. plantarum, L. acidophilus, and L. delbrueckii subspecies bulgaricus), 3 strains of Bifidobacterium (B. longum, B. breve, and B. infantis), and 1 strain of S. thermophilus, achieving a higher remission rate than control group. However, the treatment of IBD with probiotics in humans remains controversial, as various clinical studies have failed to demonstrate significant improvements in these patients. For instance, Probio-Tec AB-25 (L. acidophilus La-5 and B. animalis subsp. lactis BB-12) showed no advantage over placebo in maintaining remission in patients with left-sided ulcerative colitis, and B. breve strain Yakult had no effect on time to relapse in ulcerative colitis patients. This is not surprising, considering that IBD is a multifactorial disease and the microbiota is potentially influenced by diet, antibiotics, and other factors.”.

Additionally, in the mentioned section we already had this phrase regarding SIBO: page 8, lines 270-277 “Probiotics are emerging as a new approach to treat SIBO together with antibiotics; in fact, a meta-analysis has reported that probiotics improve SIBO by increasing the decontamination rate, reducing hydrogen concentration, and alleviating abdominal pain. The probiotics studied included strains of Lactobacillus (such as L. casei, L. acidophilus, L. rhamnosus), Bifidobacterium (B. breve, B. longum, B. infantis), and Bacillus clausii, among others, either alone or in combination with antibiotics. However, probiotics were not effective in preventing SIBO.”.

In the section of metabolic disorders, we had already written: page 8, lines 311-314 “A systematic review of 16676 overweight and obese adults found that probiotics had a moderate effect on reducing body weight; however, these beneficial effects were only observed when probiotics were used in high doses”. Nevertheless, to provide a more detailed discussion, we have added the following lines: page 8, lines 314-317 “In fact, the authors of the mentioned review highlight that most studies used probiotic mixtures, making it hard to identify the most effective strains. Variations in intervention duration, dosage, and participant characteristics could explain the discrepancies, emphasizing the need for further research on anti-obesity strains.”.

In the section of neurological diseases we had already written: page 10, lines 372- 375 “For example, L. plantarum can reduce α-synuclein accumulation in the substantia nigra in Parkinson’s disease, while B. animalis and L. acidophilus have been shown to rescue nigral dopaminergic neurons from 1-methyl-4-phenyl-1,2,3,6-tetrahydropyridine (MPTP) and rotenone-induced neurotoxicity.”.

Furthermore, we have added an additional paragraph discussing:

  • 3 recent meta-analysis about Parkinson’s disease, Alzheimer’s disease and mild cognitive impairment where all of the studies showed positive effect on probiotic use. Anyway, we added this phrase at the end of the paragraph “One aspect to consider is that these kinds of studies apply preliminary filtering, which may exclude inconclusive trials, and include studies in which the probiotics used are not the same or consist of various genera and species, despite reaching a uniform conclusion.”
  • 1 recent meta-analysis about anxiety and probiotics, where non-significant decrease of anxiety was found in humans. “However, a recent meta-analysis revealed that only a limited number of probiotic species are associated with a reduction in anxiety in animals. This was observed, for instance, with L. rhamnosus. Nevertheless, in human studies, this strain did not produce significant improvements in anxiety symptoms.”
  • Another application of probiotics in relation to neurobiology and postoperative contexts, in which we highlighted the limited sample sizes often found in these types of studies. Therefore, the observed results should be interpreted with caution: page 10, lines 408

Finally, in ‘Dysbiosis and probiotics in the immune system”, we have included additional information about randomized controlled trials in rheumatoid arthritis, ulcerative colitis, and multiple sclerosis including a meta-analysis, indicating that “restoring the microbiome or introducing beneficial microorganisms that help reestablish immunological balance may alleviate symptoms and improve the prognosis of these conditions”.

  1. Clinical applicability and practical implications remain uncertain due to contradictory findings and inconsistent evidence presented throughout.

We agree with your perspective. In this new version, we have included and clarified several clinical trials within each section discussing the existing controversies and highlighting clinical studies in which the benefits of probiotics were not observed.

  1. Structural issues, including repetitive content and poor narrative organization, significantly compromise the manuscript's overall readability and coherence.

We have reviewed the entire manuscript, removed repetitive content, simplified sections that were overly redundant, and improved the overall narrative. We have worked diligently over the past two weeks to substantially improve the manuscript, as reflected in the extensive revisions made throughout the text. All changes have been clearly marked to facilitate the review process and to highlight the significant updates implemented in response to the reviewers’ valuable feedback.

Reviewer 4 Report

Comments and Suggestions for Authors

The article under review focuses on the analysis and systematization of experimental studies and reviews on probiotics and their practical use. The authors analyzed the most common pathologies resulting from an imbalance in the gut microbiota, as well as detailed the most important and known gut probiotics, their functions, and the mechanisms of action in relation to these conditions. The work is structured and contains a large list of references. The available data are illustrated in a single figure and table. The figure is demonstrative, while the table is of a reference and informational nature. The authors systematized and presented the bulk of the experimental data in this table. Considering that any review of literary information is aimed primarily at the correct interpretation of this information, my main comments concern checking the accuracy of the citations and references used. 

  1. P.2, L. 45-47: The authors write: “In other organs, such as the intestine, dysbiosis can entail significantly more negative aspects, including … an increased risk of cardiovascular diseases [14]”, however, in the abstract of ref.14 we found that “In this Review, we examine the link between the oral microbiome and CVD, specifically coronary artery disease, stroke, peripheral artery disease, heart failure, infective endocarditis and rheumatic heart disease.” The citation should be checked carefully.
  2. P. 2, L. 76-81, and also further: The authors used an old taxonomy of bacteria in the text of the article; therefore, it is necessary to carefully proofread and make corrections in accordance with the current taxonomy. See https://www.bacterio.net/ and Oren A, Garrity GM. Valid publication of the names of forty-two phyla of prokaryotes. Int J Syst Evol Microbiol 2021; 71:5056.

The authors should correct the bacterial phylum and genera names of Lactobacillus and Propionibacterium throughout the text to those currently in accordance with current taxonomy. Authors may choose one of two options: (1) Firmicutes (current name Bacillota) or (2) Bacillota (blast name, homotypic synonym Firmicutes) for the phylum names, while using the current name vs homotypic synonym status for the genus-species names Lactobacillus and Propionibacterium.

  1. P. 3, L. 109: check reference 38. The authors write "Furthermore, most bacteria of these genera also decrease intestinal pH through organic acid production, thereby inhibiting the growth of pathogenic bacteria, which usually prefer a neutral pH [38]." In this case, the authors refer to species belonging to the genera Lactobacillus and Bifidobacterium.  However, in the cited publication, according to the methods and main results, it is said about "Fifteen segments of ileum from 14 guinea-pigs were infused intraluminally with lactic acid 0.05-2% v/v or equivalent amounts of sodium lactate or lactulose, acetic acid 0.4 or 0.5 % v/v, HCl O-012 or 0-5 % w/v, or control infusion of bicarbonate-free Krebs solution." However, no bacterial strains were tested.
  2. P. 3, L. 112-116: The authors provide a very narrow range of characteristics for potential probiotic microorganisms, citing publications from 2010 (40) and 2014 (41). I recommend that the authors provide more up-to-date information on this topic.
  3. Key comments on Table 1:

5.1. Provide correct information on the taxonomy of lactobacilli: L. brevis, correct name Levilactobacillus brevis; L. casei, correct name Lacticaseibacillus casei, and for the following lactobacillus species: fermentum, paracasei, plantarum, reuteri, rhamnosus, salicinius, and salivarius.

5.2. Lactobacillus acidophilus: check in ref. 71 the information on “Mechanisms of action / Benefits”, add data.

5.3. Lactobacillus brevis, Lactobacillus casei: check in ref. 47 the information on “Mechanisms of action / Benefits”, add data.

5.4. Lactobacillus casei: check in ref. 72 the information on “Mechanisms of action / Benefits”, add data. 

5.5. Missing information on Lactobacillus delbrueckii subsp. lactis, which is in ref. 36, check the information on “Mechanisms of action / Benefits”, add data.

5.6. Lactobacillus gasseri: check in refs. 48 and 62 the information on “Mechanisms of action / Benefits”, add data.

5.7. Lactobacillus helveticus: check in ref. 105 the information on “Mechanisms of action / Benefits”, add data. 

5.8. Lactobacillus johnsonii: check in ref. 48 the information on “Mechanisms of action / Benefits”, add data.

5.9. Lactobacillus paracasei: check in refs. 48 and 80 the information on “Mechanisms of action / Benefits”, add data. Check the accuracy of the citation ref. 71.

5.10. Missing information on Lactiplantibacillus pentosus, which is in ref. 48, check the information on “Mechanisms of action / Benefits”, add data.

5.11. Lactobacillus plantarum: check in refs. 48 and 71 the information on “Mechanisms of action / Benefits”, add data. 

5.12. Lactobacillus reuteri: check in ref. 72 the information on “Mechanisms of action / Benefits”, add data. 

5.13. Lactobacillus rhamnosus: check in refs. 80 and 105 the information on “Mechanisms of action / Benefits”, add data. 

5.14. Lactobacillus salivarius: check in refs. 25, 47 and 72 the information on “Mechanisms of action / Benefits”, add data. 

5.15. Provide correct taxonomic information: Bifidobacterium infantis and Bifidobacterium lactis.

5.16. Carefully check the correctness of the citation of ref. 36 for all species of bifidobacteria. These species were not mentioned in the cited reference. Also clarify the data in ref. 71 for B. infantis as well as in ref. 72 for B. bifidum and B. dentium.

5.17. Bifidobacterium adolescentis: check in ref. 107 the information on “Mechanisms of action / Benefits”, add data. 

5.18. Bifidobacterium animalis: check in ref. 105 the information on “Mechanisms of action / Benefits”, add data. 

5.19. Bifidobacterium breve: check in refs. 71, 78, and 105 the information on “Mechanisms of action / Benefits”, add data. 

5.20. Bifidobacterium bifidum: check in ref. 78 the information on “Mechanisms of action / Benefits”, add data. 

5.21. Bifidobacterium infantis: check in ref. 78 the information on “Mechanisms of action / Benefits”, add data. 

5.22. Bifidobacterium lactis: check in refs. 46, 71, 72 the information on “Mechanisms of action / Benefits”, add data. 

5.23. Bifidobacterium longum: check in refs. 105 and 107 the information on “Mechanisms of action / Benefits”, add data. 

5.24. Enterococcus faecalis: check in ref. 62 the information on “Mechanisms of action / Benefits”, add data. 

5.25. Lactococcus lactis ssp. cremoris: check in ref. 105 the information on “Mechanisms of action / Benefits”, add data. 

5.26. Lactococcus lactis ssp. lactis: check in ref. 47 the information on “Mechanisms of action / Benefits”, add data. 

5.27. Streptococcus «salibarius» replace with «salivarius».

5.28. Streptococcus boulardii: check in ref. 48 the information on “Mechanisms of action / Benefits”, add data. 

5.29. Provide correct taxonomic information for Propionibacterium acidipropionici, P. jensenii, and P. thoenii

5.30. Propionibacterium acidipropionici and P. freudenreichii: check in ref. 107 the information on “Mechanisms of action / Benefits”, add data. 

5.31. In the context of the presented table, I would like to note that the authors should coordinate citation of the following refs: 71, 105, and 107. In original articles, the authors describe in detail the composition of the strain mixture or a list of the strains tested.

  1. P. 6, L. 216: First and only mention of F. prausnitzii – give the full name of the microorganism.
  2. P. 9, L. 360: First and only mention of L. sporogenes, C. butyricum, and B. mesentericus – give the full name of the microorganisms.

Author Response

The article under review focuses on the analysis and systematization of experimental studies and reviews on probiotics and their practical use. The authors analyzed the most common pathologies resulting from an imbalance in the gut microbiota, as well as detailed the most important and known gut probiotics, their functions, and the mechanisms of action in relation to these conditions. The work is structured and contains a large list of references. The available data are illustrated in a single figure and table. The figure is demonstrative, while the table is of a reference and informational nature. The authors systematized and presented the bulk of the experimental data in this table. Considering that any review of literary information is aimed primarily at the correct interpretation of this information, my main comments concern checking the accuracy of the citations and references used. 

  1. P.2, L. 45-47: The authors write: “In other organs, such as the intestine, dysbiosis can entail significantly more negative aspects, including … an increased risk of cardiovascular diseases [14]”, however, in the abstract of ref.14 we found that “In this Review, we examine the link between the oral microbiome and CVD, specifically coronary artery disease, stroke, peripheral artery disease, heart failure, infective endocarditis and rheumatic heart disease.” The citation should be checked carefully.

First of all, we would like to sincerely thank Reviewer 4 for the thorough analysis of our review and their highly constructive comments, which have significantly helped us to improve the manuscript.

We agree that reference [14] was not appropriate in this context and have replaced it with a more suitable citation: Lau, K.; Srivatsav, V.; Rizwan, A.; Nashed, A.; Liu, R.; Shen, R.; Akhtar, M. Bridging the Gap between Gut Microbial Dysbiosis and Cardiovascular Diseases. Nutrients 2017, 9, doi:10.3390/NU9080859.

  1. P. 2, L. 76-81, and also further: The authors used an old taxonomy of bacteria in the text of the article; therefore, it is necessary to carefully proofread and make corrections in accordance with the current taxonomy. See https://www.bacterio.net/ and Oren A, Garrity GM. Valid publication of the names of forty-two phyla of prokaryotes. Int J Syst Evol Microbiol 2021; 71:5056.

The authors should correct the bacterial phylum and genera names of Lactobacillus and Propionibacterium throughout the text to those currently in accordance with current taxonomy. Authors may choose one of two options: (1) Firmicutes (current name Bacillota) or (2) Bacillota (blast name, homotypic synonym Firmicutes) for the phylum names, while using the current name vs homotypic synonym status for the genus-species names Lactobacillus and Propionibacterium.

Thank you for this important comment. We have revised the manuscript to ensure consistency with the current bacterial taxonomy. The names of Lactobacillus, Propionibacterium, and the phylum Firmicutes/Bacillota have been updated accordingly throughout the text. Also, we have updated Bacteroidetes to Bacteroidota, Actinobacteria to Actinomycetota, Proteobacteria to Pseudomonadota, Verrucomicrobia to Verrucomicrobiota, and Fusobacteria to Fusobacteriota.

  1. P. 3, L. 109: check reference 38. The authors write "Furthermore, most bacteria of these genera also decrease intestinal pH through organic acid production, thereby inhibiting the growth of pathogenic bacteria, which usually prefer a neutral pH [38]." In this case, the authors refer to species belonging to the genera Lactobacillus and Bifidobacterium. However, in the cited publication, according to the methods and main results, it is said about "Fifteen segments of ileum from 14 guinea-pigs were infused intraluminally with lactic acid 0.05-2% v/v or equivalent amounts of sodium lactate or lactulose, acetic acid 0.4 or 0.5 % v/v, HCl O-012 or 0-5 % w/v, or control infusion of bicarbonate-free Krebs solution." However, no bacterial strains were tested.

We have corrected the sentence and added two new references which are more appropriate to support the role of both Lactobacillus and Bifidobacterium in lowering intestinal pH and inhibiting pathogens: page 5, lines 131-134 “The mechanisms of action of probiotic bacteria are diverse. Many of these microorganisms produce metabolic byproducts such as acetic, propionic, and lactic acids, which contribute to a reduction in intestinal pH, thereby inhibiting the growth of pathogenic bacteria that usually prefer a neutral pH.”.

Fayol-Messaoudi, D.; Berger, C.N.; Coconnier-Polter, M.H.; Liévin-Le Moal, V.; Servin, A.L. PH-, Lactic Acid-, and Non-Lactic Acid-Dependent Activities of Probiotic Lactobacilli against Salmonella Enterica Serovar Typhimurium. Appl Environ Microbiol 2005, 71, 6008–6013, doi:10.1128/AEM.71.10.6008-6013.2005.

Yamamura, R.; Inoue, K.Y.; Nishino, K.; Yamasaki, S. Intestinal and Fecal PH in Human Health. Frontiers in Microbiomes 2023, 2, 1192316, doi:10.3389/FRMBI.2023.1192316.

  1. P. 3, L. 112-116: The authors provide a very narrow range of characteristics for potential probiotic microorganisms, citing publications from 2010 (40) and 2014 (41). I recommend that the authors provide more up-to-date information on this topic.

Thank you for the observation. We have updated this section by incorporating two more recent references that reflect current criteria for the selection of probiotic strains: page 3, line 108.

Maftei, N.M.; Raileanu, C.R.; Balta, A.A.; Ambrose, L.; Boev, M.; Marin, D.B.; Lisa, E.L. The Potential Impact of Probiotics on Human Health: An Update on Their Health-Promoting Properties. Microorganisms 2024, 12, doi:10.3390/MICROORGANISMS12020234.

Ayyash, M.M.; Abdalla, A.K.; AlKalbani, N.S.; Baig, M.A.; Turner, M.S.; Liu, S.Q.; Shah, N.P. Invited Review: Characterization of New Probiotics from Dairy and Nondairy Products-Insights into Acid Tolerance, Bile Metabolism and Tolerance, and Adhesion Capability. J Dairy Sci 2021, 104, 8363–8379, doi:10.3168/JDS.2021-20398.

  1. Key comments on Table 1:

5.1. Provide correct information on the taxonomy of lactobacilli: L. brevis, correct name Levilactobacillus brevis; L. casei, correct name Lacticaseibacillus casei, and for the following lactobacillus species: fermentum, paracasei, plantarum, reuteri, rhamnosus, salicinius, and salivarius.

We have added the new taxonomy nomenclature in parentheses and retained the former one to facilitate the tracking and grouping of the original table.

5.2. Lactobacillus acidophilus: check in ref. 71 the information on “Mechanisms of action / Benefits”, add data.

We have added ref. 71 for L. acidophilus and checked all Mechanisms of action/Benefits of this microorganism.

5.3. Lactobacillus brevis, Lactobacillus casei: check in ref. 47 the information on “Mechanisms of action / Benefits”, add data.

We have carefully checked both microorganisms in ref. 47 and have added the missing Mechanism of action/Benefit in the Table: 17 “Prevention/improve of neural diseases or disorders”.

5.4. Lactobacillus casei: check in ref. 72 the information on “Mechanisms of action / Benefits”, add data. 

We have carefully checked L. casei in the ref. 72 and have added the missing Mechanism of action/Benefit in the Table: 14 “Buccal health improvement”.

5.5. Missing information on Lactobacillus delbrueckii subsp. lactis, which is in ref. 36, check the information on “Mechanisms of action / Benefits”, add data.

We have carefully checked L. delbrueckii subsp. lactis in ref. 36 and have added this microorganism with its Mechanisms of action/Benefits in the Table: 1 “Decrease intestinal pH and regulate microflora/prevention of infections”, 2 “Produce metabolites/enzymes that improve nutrition”, 4 “Modulate immunity through anti-inflammatory or pro-inflammatory cytokines”, 7 “Reduce oxidative stress/show antioxidant activity”, and 11 “Positive effect against metabolic diseases like obesity or diabetes”. We have now included ref. 36 for L. delbrueckii subsp. lactis, along with an additional relevant reference.

5.6. Lactobacillus gasseri: check in refs. 48 and 62 the information on “Mechanisms of action / Benefits”, add data.

We have carefully checked L. gasseri in refs. 48 and 62 and have added both citations for this microorganism in the Table. Additionally, we have included the missing Mechanisms of action/Benefits: 2 “Produce metabolites/vitamins/enzymes that improve nutrition” and 4 “Modulate immunity through anti-inflammatory or pro-inflammatory cytokines”.

5.7. Lactobacillus helveticus: check in ref. 105 the information on “Mechanisms of action / Benefits”, add data. 

We have carefully checked L. helveticus in ref. 105 and have added the citation for this microorganism in the Table.

5.8. Lactobacillus johnsonii: check in ref. 48 the information on “Mechanisms of action / Benefits”, add data.

We have carefully checked L. johnsonii in ref. 48 and have added the citation for this microorganism in the Table.

5.9. Lactobacillus paracasei: check in refs. 48 and 80 the information on “Mechanisms of action / Benefits”, add data. Check the accuracy of the citation ref. 71.

We have carefully checked L. paracasei in refs. 48 and 80 and have added both citations for this microorganism in the Table. Additionally, we have removed ref. 71.

5.10. Missing information on Lactiplantibacillus pentosus, which is in ref. 48, check the information on “Mechanisms of action / Benefits”, add data.

We have added L. pentosus and its mechanisms of action/benefits in the Table, supported by ref. 48 and some other relevant citations.

5.11. Lactobacillus plantarum: check in refs. 48 and 71 the information on “Mechanisms of action / Benefits”, add data. 

We have carefully checked L. plantarum in refs. 48 and 71 and have added both citations for this microorganism in the Table. Additionally, we have included the missing Mechanisms of action/Benefits: 8 “Control cholesterol or lipid levels in blood” and 11 “Positive effect against metabolic diseases like obesity or diabetes”.

5.12. Lactobacillus reuteri: check in ref. 72 the information on “Mechanisms of action / Benefits”, add data. 

We have carefully checked L. reuteri in ref. 72 and have added the citation for this microorganism in the Table, as well as the missing Mechanism of action/Benefit: 15 “Skin diseases improvement/atopic dermatitis treatment”.

5.13. Lactobacillus rhamnosus: check in refs. 80 and 105 the information on “Mechanisms of action / Benefits”, add data. 

We have carefully checked L. rhamnosus in refs. 80 and 105 and have added both citations for this microorganism in the Table.

5.14. Lactobacillus salivarius: check in refs. 25, 47 and 72 the information on “Mechanisms of action / Benefits”, add data.

We have carefully checked L. salivarius in refs. 25, 47, and 72 and have added these citations in the Table. Additionally, we have included the missing Mechanism of action/Benefit: 2 “Produce metabolites/vitamins/enzymes that improve nutrition”.

5.15. Provide correct taxonomic information: Bifidobacterium infantis and Bifidobacterium lactis.

We have modified the taxonominc information of both microorganisms in the Table. Now: B. animalis ssp. lactis and B. longum spp. infantis.

5.16. Carefully check the correctness of the citation of ref. 36 for all species of bifidobacteria. These species were not mentioned in the cited reference. Also clarify the data in ref. 71 for B. infantis as well as in ref. 72 for B. bifidum and B. dentium.

We have removed the ref. 36 in B. pseudocatenulatum, B. thermophilum, B. animalis ssp. lactis, B. animalis, and B. adolescentis. In addition, we have removed ref.71 for B. infantis and ref. 72 for B. bifidum and B. dentium.

5.17. Bifidobacterium adolescentis: check in ref. 107 the information on “Mechanisms of action / Benefits”, add data. 

We have carefully checked B. adolescentis in ref. 107 and have added the citation for this microorganism in the Table. Additionally, we have included the missing Mechanism of action/Benefit: 7 “Reduce oxidative stress/present antioxidant activity”.

5.18. Bifidobacterium animalis: check in ref. 105 the information on “Mechanisms of action / Benefits”, add data. 

We have carefully checked B. animalis in ref. 105 and have added the citation for this microorganism in the Table.

5.19. Bifidobacterium breve: check in refs. 71, 78, and 105 the information on “Mechanisms of action / Benefits”, add data. 

We have carefully checked B. breve in refs. 71, 78, and 105 and have added these citations for this microorganism in the Table. Additionally, we have included the missing Mechanism of action/Benefit: 7 “Reduce oxidative stress/present antioxidant activity”.

5.20. Bifidobacterium bifidum: check in ref. 78 the information on “Mechanisms of action / Benefits”, add data. 

We have carefully checked B. bifidum in ref. 78 and have added the citation for this microorganism in the Table.

5.21. Bifidobacterium infantis: check in ref. 78 the information on “Mechanisms of action / Benefits”, add data. 

We have carefully checked B. infantis in ref. 78 and have added the citation for this microorganism in the Table. Additionally, we have included the missing Mechanisms of action/Benefits: 4 “Modulate immunity through anti-inflammatory or pro-inflammatory cytokines” and 7 “Reduce oxidative stress/present antioxidant activity”.

5.22. Bifidobacterium lactis: check in refs. 46, 71, 72 the information on “Mechanisms of action / Benefits”, add data.

We have carefully checked B. lactis in refs. 46, 71, and 72 and have added these citations for this microorganism in the Table. Additionally, we have included the missing Mechanisms of action/Benefits: 6 “Beneficial role against intestinal diseases like inflammatory intestinal injury, ulcerative colitis, decline crypt depth” and 14 “Buccal health improvement”.

5.23. Bifidobacterium longum: check in refs. 105 and 107 the information on “Mechanisms of action / Benefits”, add data.

We have carefully checked B. longum in refs. 105 and 107 and have added these citations for this microorganism in the Table. and checked the mechanisms of action. Additionally, we have included the missing Mechanism of action/Benefit: 2 “Produce metabolites/vitamins/enzymes that improve nutrition”.

5.24. Enterococcus faecalis: check in ref. 62 the information on “Mechanisms of action / Benefits”, add data. 

We have carefully checked E. faecalis in ref. 62 and have only found that Lactobacillus inhibits the growth of E. faecalis.

5.25. Lactococcus lactis ssp. cremoris: check in ref. 105 the information on “Mechanisms of action / Benefits”, add data. 

We have carefully checked L. lactis ssp. cremoris in ref. 105 and have added the citation for this microorganism in the Table.

5.26. Lactococcus lactis ssp. lactis: check in ref. 47 the information on “Mechanisms of action / Benefits”, add data. 

We have carefully checked L. lactis ssp. lactis in ref. 47 and have added the citation for this microorganism in the Table. Additionally, we have included the missing Mechanism of action/Benefit: 17 “Prevention/improve of neural diseases or disorders”.

5.27. Streptococcus «salibarius» replace with «salivarius».

Done.

5.28. Streptococcus boulardii: check in ref. 48 the information on “Mechanisms of action / Benefits”, add data. 

Thank you for your comment. We believe there may have been a confusion, as ref. 48 discusses Saccharomyces boulardii (S. boulardii), not Spretococcus boulardii. As S. boulardii is a fungal probiotic, it was excluded from our analysis, which focused specifically on bacterial species. While S. boulardii is indeed a well-established and widely used probiotic, including it would have required the consideration of additional fungal strains, thereby expanding the scope of the review beyond its intended focus.

5.29. Provide correct taxonomic information for Propionibacterium acidipropionici, P. jensenii, and P. thoenii

Thank you for your comment. We have added the current taxonomic names of these microorganisms in the Table: Acidipropionibacterium acidipropionici, Acidipropionibacterium jensenii, and Acidipropionibacterium thoenii.

5.30. Propionibacterium acidipropionici and P. freudenreichii: check in ref. 107 the information on “Mechanisms of action / Benefits”, add data. 

We have added ref. 107 for P. acidipropionici in the Table. Additionally, we have included the missing Mechanism of action/Benefit in both microorganisms: 7 “reduce oxidative stress/present antioxidant activity”.

5.31. In the context of the presented table, I would like to note that the authors should coordinate citation of the following refs: 71, 105, and 107. In original articles, the authors describe in detail the composition of the strain mixture or a list of the strains tested.

Since ref. 71 employed a mixture of Lactobacillus acidophilus, Bifidobacterium lactis, Lactobacillus plantarum, and Bifidobacterium breve, we have corrected the use of the ref., removing it from other microorganisms in the Table.

The use of refs. 105 and 107 has been carefully reviewed and appropriately corrected in the Table.

  1. P. 6, L. 216: First and only mention of F. prausnitzii – give the full name of the microorganism.

We have included the full name.

  1. P. 9, L. 360: First and only mention of L. sporogenes, C. butyricum, and B. mesentericus – give the full name of the microorganisms.

We have included the full names.

Round 2

Reviewer 2 Report

Comments and Suggestions for Authors

The authors fully addressed all my comments and significantly improved the quality of the manuscript.

Author Response

Thank you very much for your contribution.

Reviewer 3 Report

Comments and Suggestions for Authors

You can add this article to improve visibility: 1. https://www.mdpi.com/2624-5647/5/3/28

Author Response

Thank you for your suggestion. We have added some lines addressing the mentioned article regarding the association between COVID-19 and the gut microbiota: page 2, lines 51-56.

“In fact, the relevance of gut microbiota imbalances extends beyond chronic and metabolic conditions. Emerging evidence also highlights its role in acute infectious diseases, such as COVID-19. SARS-CoV-2 infection has been associated with significant alterations in gut microbial composition, reduced diversity, and increased intestinal inflammation, suggesting a broader systemic impact of dysbiosis in disease severity and immune response modulation”.

Reviewer 4 Report

Comments and Suggestions for Authors

I am satisfied with the corrections the authors made to the text.

Author Response

Thank you very much for your corrections.